# The Rapid Adaptation and Optimisation of a Digital Behaviour-Change Intervention to Reduce the Spread of COVID-19 in Schools

**DOI:** 10.3390/ijerph19116731

**Published:** 2022-05-31

**Authors:** Georgia Treneman-Evans, Becky Ali, James Denison-Day, Tara Clegg, Lucy Yardley, Sarah Denford, Rosie Essery

**Affiliations:** 1Bristol Medical School, University of Bristol, Bristol BS8 1QU, UK; j.l.denison-day@soton.ac.uk (J.D.-D.); js18722@bristol.ac.uk (T.C.); lucy.yardley@bristol.ac.uk (L.Y.); sarah.denford@bristol.ac.uk (S.D.); r.a.essery@soton.ac.uk (R.E.); 2Primary Care Research Centre, University of Southampton, Southampton SO16 5ST, UK

**Keywords:** behaviour change, digital intervention, COVID-19, school

## Abstract

The rapid transmission of COVID-19 in school communities has been a major concern. To ensure that mitigation systems were in place and support was available, a digital intervention to encourage and facilitate infection-control behaviours was rapidly adapted and optimised for implementation as a whole-school intervention. Using the person-based approach, ‘Germ Defence’ was iteratively adapted, guided by relevant literature, co-production with Patient and Public Involvement representatives, and think-aloud interviews with forty-five school students, staff, and parents. Suggested infection-control behaviours deemed feasible and acceptable by the majority of participants included handwashing/hand-sanitising and wearing a face covering in certain contexts, such as crowded public spaces. Promoting a sense of collective responsibility was reported to increase motivation for the adoption of these behaviours. However, acceptability and willingness to implement recommended behaviours seemed to be influenced by participants’ perceptions of risk. Barriers to the implementation of recommended behaviours in school and at home primarily related to childcare needs and physical space. We conclude that it was possible to rapidly adapt Germ Defence to provide an acceptable resource to help mitigate against infection transmission within and from school settings. Adapted content was considered acceptable, persuasive, and accessible.

## 1. Introduction

Due to COVID-19, primary and secondary schools across the UK experienced repeated disruptions throughout 2020 and 2021, affecting the education of 8.9 million students [1]. Whilst it was important for students to return to in-school education as soon as safely possible, the rapid transmission of COVID-19 was a major concern. The high number of contacts within a school setting, coupled with the potential occurrence of asymptomatic cases [2] and low levels of vaccination [3], gave rise to the possibility of rapid onward transmission to family members and the wider community. Therefore, it was critical to ensure that systems were in place and support was available to help mitigate against infection and transmission within and from school settings.

Throughout the pandemic, Government guidance advocated that schools implement infection-control measures to protect staff, students, and their household contacts. At various times, these measures included wearing a face covering, social distancing, handwashing and/or sanitising, and increasing efforts to ensure that spaces were well-ventilated and clean [4]. However, these measures were not always perceived as feasible or sustainable [5], and frequently changing guidance could leave school staff and students unsure of the best way to reduce transmission of COVID-19 [6]. As such, an accessible, persuasive, and practicable intervention was required to support school staff, students, and parents in adopting of behaviours to reduce the transmission of COVID-19, as well as potential future viral outbreaks.

Interventions for school settings to help encourage hand-hygiene and environmental-disinfection behaviours already exist [7]. A recent systematic review concluded that interventions that include behaviour-change techniques targeting multiple theoretical approaches, and facilitating the development of the capability, motivation, and opportunity to carry out behaviours appear to be associated with the best outcomes [7,8,9]. However, many existing interventions for school-based settings do not adequately address the behavioural and logistical barriers experienced by this population, and further exploratory research is required [7]. Furthermore, as very few existing interventions for school settings focus on hand-hygiene and environmental-disinfecting behaviours simultaneously (and to our knowledge, none have been created for pandemic-like situations), research is now needed to explore facilitators and barriers to such infection-control behaviours in the context of the COVID-19 pandemic.

Previous research has suggested that school-based interventions are most effective when integrated into daily practice and school culture, seeking to engage all staff, reinforce skills outside of the classroom, support parental engagement, and coordinate work with outside services [10]. Consequently, it is important to understand the experiences, perceptions, and perceived barriers to relevant behaviours in the context of COVID-19 for relevant groups of individuals (e.g., school staff, students, and parents), in order to ensure that a whole-school intervention can be developed to address such issues.

A digital behaviour-change intervention, ‘Germ Defence’, exists as a website to help users to understand the measures to take and when to take them in order to avoid infection. These measures include hand washing and ventilating rooms, and the intervention guides users through behaviour-change techniques to adopt better health habits and find ways to solve any barriers [11]. Germ Defence has previously been shown to increase handwashing in the home and reduce the transmission of respiratory tract infections in a randomised controlled trial of more than 20,000 adults [11]. Germ Defence was originally developed using the person-based approach, drawing on in-depth qualitative understandings of target users, alongside theory and evidence, to develop detailed understandings of how to overcome the behavioural barriers users might encounter in engaging with the target behaviours [12]. The intervention incorporated information, personalised goal setting, environmental prompting, and persuasive communications to increase motivation. A process evaluation demonstrated Germ Defence was effective for both men and women, for older and younger adults, and for people with varying levels of education [13]. In early 2020, Germ Defence was rapidly adapted for use during the COVID-19 pandemic by a team of medical, public health and behaviour-change experts, and public contributors [14,15]. It was then disseminated through multiple pathways, including public health and primary care networks, national and local press, television coverage, and social media [14,15]. Over 600,000 people have now used this version of Germ Defence. Research was needed to rapidly explore how Germ Defence could be adapted for use by parents, students, and teachers in both school and home settings during the COVID-19 pandemic. 

The aim of the present study was to use the person-based approach to rapidly adapt and optimise the existing Germ Defence for implementation in a school setting for use by students, school staff and parents as a whole-school intervention. 

Key objectives were: To explore the experiences of students, school staff, and parents and their perceptions of behavioural recommendations relevant to COVID-19.To explore reactions to—and beliefs about—the content, structure, and format of Germ Defence.To use these insights to adapt Germ Defence in line with user experience, and as a result, develop an intervention that is acceptable, persuasive, and feasible amongst school communities.

## 2. Materials and Methods

This study was nested within a larger UK Research and Innovation (UKRI)-funded project investigating how to effectively reduce the transmission of COVID-19 in schools (Coronavirus Mapping and Mitigation in Schools—CoMMinS) (NIHR Project Ref: COV0591) [16]. The study followed Medical Research Council (MRC) guidance on intervention development and evaluation [17] and employed the validated person-based approach to optimise intervention content. The person-based approach involves understanding users’ contexts and their views of every aspect of the intervention [12].

The intervention, Germ Defence, was originally developed using the LifeGuide software (Markham, Canada). As part of the current body of research, this was adapted to make use of modern web technologies, with a particular focus on responsive design (layouts that will adapt to different devices), accessibility (in line with WCAG AA guidelines) and usability.

This study involved three key activities, each outlined below, and displayed in Figure 1: Collating evidence from relevant literature and input from Patient and Public Involvement (PPI) contributors to inform guiding principles and provisional intervention adaptationsIn-depth qualitative think-aloud interviews with school staff, parents, and studentsAnalysis of this qualitative data to provide further understanding of these individuals’ experiences and perceptions of the pandemic and relevant behaviours, and to inform how Germ Defence could be further adapted and optimised.


Collating evidence and PPI input to inform Guiding Principles


The initial stage of the study involved (1) a brief review of recent and relevant literature, (2) online discussion groups with PPI contributors, and (3) collation of this evidence and PPI feedback into an ‘intervention planning table’ to inform the development of guiding principles. 

The search for relevant literature was initially broad in its scope, and included: staff and students’ views of attending school during the pandemic [6]; factors affecting handwashing behaviours among young children [18,19,20]; behavioural determinants of handwashing among all ages [21]; and the effectiveness of hand hygiene interventions in educational settings [22]. Later, we focused on more specific literature to understand and inform optimal delivery formats, styles, and techniques of behaviour-change interventions previously implemented in this population [23]. This more focused research included evaluations of school-based interventions targeting physical activity, healthy eating, sexual and reproductive health, mental health, and use of tobacco, alcohol, and other drugs [24]. The team also reviewed emerging literature regarding how COVID-19 is transmitted [25], including airborne transmission [26] and the importance of indoor ventilation control [27,28]. Then-current evidence and relevant Government guidance was reviewed by the team as it emerged, in order to ensure the intervention’s content was informed by the most up-to-date evidence and regulations. 

Public involvement groups were established, and included: eight school staff members (primary and secondary) who were invited to four group meetings, nine students aged 11–18 years invited to two group meetings, and six parents of students invited to one group meeting. PPI members were identified via an existing Young Person Advisory Group (YPAG) and school staff members and parents were recruited via social media. During a series of online meetings, the groups’ discussions aimed to explore and understand the then-current infection-control practices within schools (hand washing, wearing of face coverings, social distancing (including ‘bubbles’), ventilation and cleaning), how Germ Defence might be used within schools to address challenges, and if they felt anything was missing from these infection-control practices. All PPI groups were facilitated by two members of the team (RE & SD), with support from a PPI involvement facilitator during the group meetings with students (Appendix A; document A shows an example of the topic guide used).

Data gathered in the initial phase were collated into the intervention planning table (Appendix A). The intervention planning table recorded specific discussions with PPI contributors and existing literature, and the possible implications of these in terms of potential changes to the existing intervention content and features. The intervention planning table informed the development of preliminary ‘guiding principles’ for the adapted Germ Defence intervention. Guiding principles are a core element of the person-based approach, and are a way of applying knowledge about an intervention’s target users and the context for the relevant behaviours in order to maximise the acceptability and feasibility of the intervention being developed [12]. These guiding principles informed the initial suggestions for modifying the existing intervention content, and Germ Defence was provisionally amended on the basis of this. PPI input also informed development of the qualitative interview schedule by highlighting areas where further feedback from school staff, students, and parents would be valuable.

II.In-depth qualitative think-aloud interviews

In-depth qualitative think-aloud interviews with members of school staff, parents, and students explored reactions to intervention content, as well as both barriers and facilitators to adopting infection-control behaviours. Throughout this phase, the team continued to review emerging evidence and Government guidance and incorporated this into the intervention’s content where necessary. 


**Recruitment of schools**


Seventeen schools from Bristol and the surrounding area had agreed to participate in the wider CoMMinS study [16]: eight primary, seven secondary, and two Special Educational Needs (SEN) schools. These schools were initially contacted and invited to take part in the current study according to educational stage (primary or secondary), representation of ethnic minority students, and free school meal eligibility as a measure of socioeconomic status. This was to reach as diverse a sample of mainstream schools as possible to ensure the intervention was developed to be as accessible as possible to individuals from different backgrounds. A total of 11 primary and secondary schools were invited to participate, with staff and parents to be recruited from the former, and staff, parents, and students from the latter. Eight of these schools expressed interest in participating in this sub-study (Figure 2). 


**Recruitment of participants**


Once a school had expressed interest in the present study, the research team approached a nominated school contact via email or phone, and further details were provided. The school contact then shared a brief summary of the study and contact details for the research team with their colleagues, as well as with their parent and student community through school newsletters and emails and during tutor group periods. Interested individuals were encouraged to contact the research team directly to express their interest in taking part in an interview. Recruitment of participants via this process was slow due to schools’ limited capacity, so recruitment of parents and members of school staff was extended to advertisements via social media. Participants were purposively sampled to ensure, as far as possible, they represented a wide range of users e.g., of school staff, parents, and students. 

Primary and SEN students were excluded from the current study, as PPI feedback suggested primary-aged children and SEN students would be unlikely to engage with the intervention independently, and that their engagement would be more probable under the supervision of a parent, carer, or school staff member. Creating a more accessible version of Germ Defence for children within these groups was outside the scope of the current study. 


**Procedure**


Interviews were conducted between April and July 2021 at a time when England was experiencing its third national lockdown and plans for lifting restrictions were gradually being introduced. Just prior to this, the Government had announced a planned return for primary and secondary school students in England following almost 12 months of repeated disruptions. At the time of the study, vaccine roll-out to the majority of adults was well underway, but under 18s were not eligible to receive COVID-19 vaccinations. 

Participants who expressed their interest by emailing the research team were provided with an age/role-appropriate participant information sheet, and a convenient time for an interview was arranged. Informed verbal consent and assent were taken (and recorded separately) before the interview began, with parents providing consent for children under 16. All interviews were conducted by two members of the team (BA & GT) via telephone or video call and were audio recorded. Face-to-face interviewing was not possible due to the social-distancing restrictions enforced during the pandemic. Participants were asked to find a quiet room with access to a laptop, tablet, smartphone, or computer. The interviewer reminded participants of the interview’s purpose and explained the session. Participants were given the option to either ‘screen-share’ or to talk through the intervention pages as they used them so the interviewer could follow their journey. Participants were given an opportunity to ask any questions, and they were reminded that the interview would be audio-recorded and that they could stop at any time without providing a reason. The recording commenced when the participant indicated they were happy to do so. 

At the start of the interview, participants were asked a series of open-ended questions to explore perceptions of barriers and facilitators to real-life implementation of the behaviours recommended. Informed by PPI discussions, the interview schedule (Appendix A) included questions to explore participants’ experiences of the pandemic and their perceptions of Coronavirus in general. Notably, we used the term ‘Coronavirus’ with participants as we felt it was more accessible than ‘COVID-19’. It also asked questions regarding their living situation, including whether they lived with anyone who deemed themselves to be at risk of becoming seriously ill if they were to contract Coronavirus. Participants provided feedback on the Germ Defence intervention materials using the ‘think aloud’ interview method [29] offering them the opportunity to express their immediate reactions to all aspects of the intervention content and functionality as they used it in real-time. 

Participants were then asked to use Germ Defence as they normally would and to discuss their thoughts about the information and content on each page. During this element, the interviewer used neutral prompts to encourage participants to ‘think aloud’ and elaborate on any points where necessary (e.g., ’Can you tell me a bit more about that?’), to facilitate greater understanding about how participants chose to use the intervention (e.g., ’Can you tell me what made you click on that button/choose that option?’) and/or to direct their attention to any specific elements requiring attention (e.g., ’What do think about this page? /How do you feel about this information?’). Notes were kept during the interviews to help probe for more information and verify responses at transcription. Transcripts were compared with recordings and field notes to verify credibility. Interviews lasted 29–78 min. Participants were thanked for their time with a £20 shopping voucher and were asked to complete a brief demographic survey. 

Interviews were conducted in parallel to the ‘table of changes’ analysis (described below) allowing changes to the intervention to be made iteratively, based on ongoing interview feedback. This meant that later participants were viewing progressively more optimised versions of the intervention. The process continued until the feedback became less novel and it seemed that no major issues were being identified. A total of five major iterations were made, culminating in a final, sixth version. We considered data saturation had been reached once it seemed no further important changes were required.

III.Data Analysis and identification of further changes required

Two main approaches to data analysis were taken and occurred in parallel: Thematic analysis allowed us to gain an in-depth understanding of users’ experiences of the pandemic and perceptions of relevant behaviours in order to inform later refinement of intervention content and messages.Table of changes analysis aimed to provide a rapid understanding of individuals’ reactions to, and perceptions of, the intervention content to identify necessary alterations and their relative importance. All interviews were transcribed verbatim with all identifiable data removed.


**Thematic Analysis**


Interview data were thematically analysed [30], facilitated by NVivo software (Doncaster, Australia). One member of the team (BA) read through the 45 transcripts and coded them inductively, line by line. Themes derived from this initial process appeared reflective of the guiding principles, and so the remainder of the data were coded using the guiding principles as a coding framework [31]. A second team member (GT) then independently coded the data to ensure scrutiny and rigour of the coding process [32]. Discrepancies were discussed and further reviewed by the team to ensure coherence within and across codes [33].


**Table of Changes Analysis**


Positive and negative comments from the interviews about specific intervention content, features, or formatting were collated within the table of changes. Using a predefined framework, possible targets for change and the reasons for this potential change were documented. For example, criteria informing recommendations for changes included whether the change was deemed important for behaviour change to occur, whether it was easy and uncontroversial to implement, whether it was supported by repeated feedback, whether it was supported by experience (PPI discussions or existing literature), and/or whether it did not contradict the guiding principles. The prioritisation of changes was coded using the MoSCoW prioritisation hierarchy (Must have, Should have, Could have, Would like). All possible changes were discussed within the team. Modifications were made to the intervention content if it was agreed that they would enhance Germ Defence’s usability, acceptability, and likelihood of encouraging behaviour change; this process increased transparency and facilitated rapid actioning of required changes. 

## 3. Results

### 3.1. Collating Evidence and PPI Input to Inform Guiding Principles

A brief review of fifteen published and relevant articles identified key messages, including: evidence that ventilation, face coverings, handwashing, and social distancing measures successfully reduce the risk of secondary infection within a household/classroom [34]; and the need for interventions to be persuasive as well as instructional [18]. These key messages, alongside PPI contributions, were collated and summarised (see intervention planning table, Appendix A), and used to develop guiding principles. Each guiding principle comprises: a design objective that outlines a user or context-specific need, issue, or challenge identified that may affect engagement with the intervention; and proposed intervention features that are expected to address the design objective (Table 1). 

Table 1 outlines the guiding principles for Germ Defence for schools, alongside contextual information about the views of parents, school staff, and students that informed each principle. These were used to inform initial adaptations to Germ Defence in line with the key features; that the content needed to be quickly and easily accessible, framed in terms of protecting others via collective efforts whilst acknowledging the difficulties of implementing the behaviours, and offering strategies to manage this. The rationale for recommended infection-control behaviours (handwashing, social distancing, wearing face coverings, and keeping one’s surroundings safe) needed to be clearly explained, and potentially confusing or contradictory advice needed to be acknowledged.

### 3.2. In-Depth Qualitative Think-Aloud Interviews 


**Participants**


Forty-five participants located around England were interviewed, consisting of (primary and secondary) school staff (*n* = 14), students aged 11+ (*n* = 15) and parents of school-aged children (*n* = 16). One member of school staff from the PPI group also took part in an interview. All other interview participants were recruited as previously described. 

Demographic data were collected from thirty-seven participants (Table 2). The remaining participants (*n* = 8) did not complete the post-interview demographic survey.

### 3.3. Data Analysis and Identification of Further Changes Required

#### 3.3.1. Thematic Analysis 

Thematic analysis of data from the think-aloud interviews provided insight into participants’ perceptions of the pandemic, perceptions of their own and others’ vulnerability, and their beliefs about infection-control behaviours. Particularly, it appeared that perceptions of risk, concern for others’ safety, and logistical and practical barriers played important roles in determining how people reacted to Germ Defence and its recommendations. Each of these three themes are discussed below, with examples from the data and discussion of how these informed further modifications to the intervention. 

The illustrative quotes from participants below are indicated as school staff (T1–T14), parents (P1–P16), and students (C1–C15).

**Theme** **1.***Perceptions of risk*. 

The perceptions of school staff, students, and parents (collectively referred to as participants hereafter) regarding risk were varied, and appeared to be informed by factors such as whether they thought they or other members of their household were at risk of serious illness from COVID-19, their beliefs about the effectiveness of strategies for preventing within-home transmission, and the implementation of the vaccination programme. In turn, perceptions of risk appeared to influence the extent to which the infection-control behaviours recommended by Germ Defence were considered feasible and/or realistic. More stringent behaviours, such as wearing a face covering or social distancing inside the home when a household member was infected, were often not considered realistic amongst those who considered themselves/their household to be at low risk: 

*I think you do need to try and stay in control of what is actually a safe thing to do and what is a practical thing to do. Particularly around the home, if you’re not having visitors in and out and it’s your own family bubble, then I think you can be relatively secure in your own home*.(P8)

*I don’t think it’s [social distancing] necessary as a household because you live together and you do everything together. I don’t think it’s the right place to say do it in your house because you have bubbles and you’re in the bubble with your house so you don’t really need to be two metres apart in your home against your family*.(C4)

This seemed to be mirrored by many students’ attitudes towards infection-control behaviours within the classroom. Students generally identified themselves as being at low risk from transmission and low risk of becoming seriously ill from COVID-19. As a result, they often appeared less willing to try to engage with particular infection-control behaviours. More stringent behaviours were only considered feasible to students if they considered themselves or those around them to be at greater risk (the wearing of face coverings was mandatory within secondary schools at the time of the research): 

*We had a debate in the tutor group about it and it was so surprising. The students did not want to wear them [face coverings]. You had probably about a third that were like, ‘well I’m doing it not just for myself but for other people and for my nan etc.’, and then there were others going, ‘I don’t see why I should have to, I’ve not got it.’…There was a lot of resistance from the students, and it was very difficult to get them to wear masks in the corridors and things*.(T14)

Participants who deemed themselves (or someone in their household) to be at high risk of becoming seriously ill if they contracted COVID-19, however, did implement some of the more stringent behaviours. When asked what sort of things they had done to try and keep themselves and others safe, participant C7 (who considered the grandparents they had recently started living with to be at high risk) explained they were taking extra precautions: 

*Well I’ve mostly been social distancing at school and everywhere, been wearing a mask when I go into shops because I’m over 11 now, and I’ve been mostly staying up in my bedroom on my computer*.(C7)

Several parents viewed their home as a space safe from virus transmission; this perceived low risk affected their adoption of certain infection-control behaviours in the home: 

*I think I obviously didn’t think that much about ‘in the house’. I think I felt the house is my safe space and that once we were indoors, we were kind of fine, but I think I should think about that more really. About spreading the virus indoors between our bubbles*.(P7)

*I don’t want to wear face coverings at home…because it feels like a safe place, because they’re horrible to wear, and because I would think that because we don’t have people in the at-risk category in our household*.(P10)

All students interviewed were more relaxed towards social distancing, wearing a face covering, and/or increasing ventilation and cleaning inside their home as opposed to when they were at school: 


*It’s your own home and you feel like you’re just safe in here*
(C12)

Participants’ perceptions of risk varied between home and school settings due to the number of people they would be in contact with. All students and school staff who discussed this issue felt the risk of contracting COVID-19 greatly increased when they were in school, particularly if others were not following the rules: 

*I think if you’re at school, you’ve got definitely a bigger risk on your shoulders because if you do something wrong or someone else does something wrong, you’re endangering quite a lot of people there…when you’re at home I mean it’s really important, but you’re endangering less people when you’re doing it at home*.(C2)

*A lot of the parents around here didn’t really take it very seriously at the start. I found myself getting a lot more anxious about it and then with the children coming in and then seeing them out and about and not following the rules, scared me even more*.(T12)

One parent echoed these concerns and believed their children were more at risk of contracting COVID-19 from school than elsewhere: 

*So they were in bubbles of eight to ten children throughout the whole of lockdown. Now they’re back in their class of 30, so that’s 30 times more risk and they’re all in three different classes so, all of a sudden, they’ve gone from having 30 families that are accessing to now there being 90 families that are accessing, potentially, so I felt quite anxious about that when they went back*. (P2)

School staff expressed concerns regarding how their own safety was at risk from some students not following recommended behaviours within school: 

*I have been aware that I’ve been at an increased risk, and working in a school, whilst I can do everything that I can to maintain my distance and follow all of the rules, it’s difficult to make sure that the students are doing the same because they don’t have the same worries and fears about it all as I do*.(T3)

The increasing roll-out of vaccinations seemed to reduce perceived risk amongst some participants who reflected on how they and others had subsequently become less rigorous in their implementation of infection-control behaviours: 

*My grandmother’s had both of hers [vaccinations]. She’s 86 and she didn’t want us at the house [before]…she had her second vaccination and she said, ‘Oh, give it two weeks,’ and then she was happy for us to come over and sit inside*.(P11)

*I think the thing is, everybody started out with amazing intentions in the first lockdown. Come back to school and we’ve all got given a million risk assessments and we’re not all being so good. Even my job share, she’s had both her vaccinations now and she doesn’t wear her mask as much*.(T8)

Overall, these findings emphasised the importance of needing to differentiate some of the behavioural recommendations made on the basis of how people perceived their risk in different circumstances. Table 3 provides detail about how the intervention was altered to address this. These findings also contributed to enhancing the messaging throughout the intervention about engaging in recommended behaviours to protect vulnerable others.

**Theme** **2.***Concerns for, and about, others*.

Participants’ accounts revealed two different types of concerns relating to others and their behaviours: (1) Firstly, a concern for others and an acknowledgement that their own behaviour could affect those around them, but also (2) sometimes as a concern about others’ behaviour, where they felt this was not always consistent with considering and protecting others. 

Many of the students who identified their household as low risk were relatively unconcerned for their own health regarding COVID-19, however, they acknowledged others more at risk: 

*[Covid-19 is] quite serious for some people but other people not so much. Like, children, you can’t really see symptoms, sometimes…it’s more serious for other people because they might have disabilities or have a disease where they’re terminally ill*. (C8)

School staff also discussed the importance of Germ Defence’s focus on collective responsibility as further persuasive motivation for their students to adopt the suggested behaviours: 

*I like the little examples because actually I don’t think they [students] realise you’re passing it on not just to who you’ve seen but also those people that other people have seen and it’s like a massive web. I really like that green box of like an example*.(T1)

*What I really liked—I know I’m jumping forwards a little bit—was where you had to choose what you do, and how often you do it, and then get you to think about it again—about one thing, or two things, that you could change to reduce your risk, and everybody around you-s risk. The thing that I really struggle with the students is, it’s not just about them, it’s about everyone else around them, and they don’t get that*.(T3)

This focus on the needs of others was also reflected upon by the majority of parents interviewed. Participant P7 felt the messages within Germ Defence regarding vaccination were an important part of sharing advice with people they care about in order to protect them: 

*I have elderly relatives who have not been vaccinated, who don’t live in the UK who are quite behind on vaccinations, but I do feel really worried and I wanna make sure that they look after themselves and know what to do to make sure—well, to try to prevent getting it*.(P7)

Whilst many participants were concerned about protecting others, several—particularly school staff—had concerns about others’ behaviours. Some school staff, including a pregnant participant quoted below, were extremely anxious about working at a school during the pandemic due to their perceived increased risk of exposure. Although this staff member felt a responsibility to be present at school, they also felt less able to keep themselves safe as they were impacted by the actions of others: 


*I love my job and these children don’t have much so I wanted to come and to do the right thing but it is hard…*



*I felt we were just quite at risk… and some of our parents don’t tell you if they’ve been sick the night before or something so that was just… the germs or the potential to catch anything is always quite high anyway…*


*With parents I didn’t feel safe and when you’re asking… they put spaces around the school and told parents where they were supposed to stand but they didn’t pay much notice. I mean, it just felt like a battle trying to keep yourself safe*.(T12)

Anxiety felt by staff regarding health behaviours at school, and how to implement ‘rules’, highlighted the importance of interventions such as Germ Defence to focus on promoting a collective responsibility for infection control, particularly as the country’s restrictions began to ease, and amid concerns that individual apathy towards such behaviours may increase: 

*I’m worried about peoples’ attitudes towards it…since the rules were relaxed people have just stopped wearing masks, stopped social distancing. I know they don’t have to anymore, but I think people have just run their course, they’re fed up with it and they don’t want to take those measures anymore, which is a shame*.(T14)

Indeed, there was evidence that Germ Defence’s messages around the importance of protecting others came through: 

*They’re good ideas [from Germ Defence] because then it’s like less—there’s less chance of people catching COVID in school. And if one person in school gets it, then not everyone in school will get it*.(C9)

*It [Germ Defence] challenged me on the idea of wearing a face covering in my home but I think I would only do that in exceptional circumstances. It got me thinking about whether I should be wearing them when I meet with people*.(P10)

The concerns of school staff, parents, and students about the impact of COVID-19 for others, and their concerns about others’ behaviour in relation to COVID-19, indicated a widely held recognition of the importance of behaving in a way that protected those around them. This emphasised the importance of our guiding principle to promote a sense of collective responsibility, and led to the strengthening of messaging around this throughout the intervention.

**Theme** **3.***Practical barriers to adopting infection-control behaviours*.

Often, participants reported barriers of a practical or logistical nature, which impeded their ability or motivation to adopt certain recommendations. These often related to childcare responsibilities or to available space or resource issues. 

Caring responsibilities towards young children often presented a barrier for parents with regard to adopting certain infection-control behaviours. For example, none of the parents interviewed felt it feasible to keep a social distance from their child, and this included isolating if they were to contract Covid-19: 

*And you’ve gotta still look after them haven’t you, and when my son had it we didn’t make him isolate away from the rest of us because mentally for them I think that would be really, yes, to put them in one room, yes too much*.(T4)

*I don’t know whether it’s feasible with a young child. … It would be very difficult for me to do it because my five-year old’s a very cuddly little boy and to tell him that he couldn’t see his mummy for seven days while she’s locked in a bedroom would be quite hard*.(P1)

*You may still want to be with your children when they eat because of the risk of choking, if there isn’t another adult in the house you live with*.(P7)

Similarly, some children questioned the feasibility of distancing within the home if they or their parents tested positive: 

*I just don’t see that it’s realistic to keep a two-metre distance from people like your mum and dad*.(C14)

Another barrier to implementing the suggested behaviours amongst staff, students, and parents included the logistical challenges of distancing associated with the amount of space within their school or home. The majority of students interviewed, for example, discussed the challenges of adopting each of the recommended behaviours when in school: facilities were not always available for handwashing using soap and water, face coverings were sometimes forgotten, opening windows meant a cold or noisy classroom, and social distancing was rarely possible in school corridors: 

*The school’s not exactly built for this [social distancing]…for one thing, there’s 215 people in my year; it’s not exactly easy, and also the corridors aren’t the thickest and then they try and split them in half to make a one-way system but then the one-way system is so thin, it’s hard to walk down… because when everyone leaves the lessons at the same time, all 215 people are in one corridor so they can’t really try and keep us two metres apart otherwise everyone’s going to be really late for their lessons*.(C6)

*We’ve just told them they need to keep their own distance because obviously classrooms aren’t built for a two-metre gap*.(T1)

*I don’t think we could do that two metres apart indoors because we live in a small flat*.(P7)

From these findings, it was clear the intervention needed to do more to help users to understand the objective of reducing rather than eliminating risk, and to address the potentially detrimental belief that if a behaviour is not done perfectly, it is not worth doing at all. This understanding helped to focus the intervention’s messaging on acknowledging the challenges of adopting certain behaviours. Subsequently, as shown in Table 3, suggestions of alternative behaviours were provided, and the wording throughout the intervention was changed to reinforce the importance of ‘doing as much as possible’.

#### 3.3.2. Table of Changes Analysis

As described above, some of the feedback identified necessary modifications to the intervention content to optimise the acceptability and persuasiveness of Germ Defence’s messages. Our table of changes analysis, conducted alongside the thematic analysis, allowed us to rapidly collate and discuss this feedback within the team to agree the alterations that would be implemented before seeking further feedback. A summary of the main changes made to the intervention throughout this process are presented below in Table 3 with illustrative quotes (an example excerpt from the table of changes is presented as Appendix A). These data informed the iterative modification of Germ Defence to reach a final version to be disseminated to schools at the start of the 2021 Autumn term.

By addressing user feedback, we rapidly adapted Germ Defence to ensure its recommendations, structure, and tone were as relevant and persuasive as possible to school staff, parents, and students. Positive feedback indicated these changes were well received. In particular, some parents were receptive to the way the adapted messaging focused on the importance of trying to enact a behaviour as best as possible: 

*It [Germ Defence] was very pragmatic about minimising touching rather than, ‘You shalt not touch,’ but if you’ve minimised, it goes back to that viral load thing really, doesn’t it? ‘You do as much as you can do, this all helps, it’s worth doing, it’s worth your inconvenience because this is how it’s going to benefit,’ and I think that’s quite good*.(P9)

Additionally, several students reflected on the ways Germ Defence explained the impact an individual’s actions can have, and how it promoted a collective responsibility by sharing stories about other people’s experiences as motivation for adopting the suggested behaviours: 


*The first paragraph is pretty good because—I don’t know—it’s like everyone can do their bit and stuff and it makes them feel involved. It’s good that it tells everyone that you don’t know what people have at home. They could have someone who’s old. Because you never know, they might live with someone who could be at risk*
(C9)

Several participants felt the intervention offered good rationales for behaviours, as well as useful prompts and reminders for those who may have relaxed their adoption of health behaviours since having the vaccination: 

*We all know about it [Covid-19] of course, but a lot of students don’t really understand how wearing a mask can make a difference, and how sanitising, washing your hands can make a difference*.(C13)

*For your family and friends, if they’ve had the vaccination and they’re doing something that maybe they shouldn’t, you could maybe just note to them ‘maybe you shouldn’t do that, or you could do this [suggested behaviour in Germ Defence]’, and if some people think they’re invincible once they’ve had the vaccine, maybe just tip them it’s not 100% effective*.(C5)

Input from school staff, students, and parents highlighted the importance of focusing on integrating the adoption of health behaviours into daily interactions and practices within the school and home setting. Praise was received for Germ Defence’s potential to ‘bridge the gap’ between home and school as a way of reducing inconsistent messages between these settings: 

*I could see this working in school, and then in a household where, for example, the parent’s primary language may not be English, actually to have children that understand this from using it at school, they may be able to communicate that to their parents, or if there is illiteracy or any other reason why the parent may not be able to take on this information themselves. Sometimes I think you can feed it through the children*.(P10)

*I like that it’s a simple thing that they [students] can do, which is to share it with their community, which is great. We [school staff] would, potentially, put it onto our school website, or into Twitter, or Instagram—or something like that—to share it with our students and parents as well*.(T3)

Following this iterative optimisation process of Germ Defence for Schools, the sixth version of the intervention was ready to be disseminated within schools. 

## 4. Discussion

This study drew on input from PPI contributors, evidence from existing literature, and think-aloud interview data to inform the rapid optimisation of Germ Defence for use as a whole-school intervention. The adapted intervention was generally considered by students, school staff, and parents to be acceptable, accessible, and persuasive. The present study provides evidence of the suitability of Germ Defence as a whole-school intervention during the COVID-19 pandemic. 

Our results suggest that infection-control behaviours promoted by Germ Defence were generally deemed feasible and acceptable by students, school staff, and parents in the context of ongoing risks from COVID-19. Whilst some behaviours were viewed as acceptable all the time, others were only viewed as acceptable in certain circumstances. For instance, wearing a face covering in crowded spaces and regularly washing or sanitising hands were viewed as realistic behaviours to implement all the time. Recent evidence has similarly indicated that regular handwashing or hand-sanitising is an acceptable behaviour during the pandemic [35,36] and our study provides support for the acceptability of handwashing in schools. 

By using the person-based approach, our data were able to highlight which of the recommended behaviours were considered unrealistic or unacceptable to implement. For example, socially distancing within small classrooms and wearing a face covering around vulnerable family members at home were only considered to be feasible in extreme circumstances. Indeed, an individual’s willingness to adopt specific behaviours appeared to be influenced by the interrelationship between risk perceptions and practical barriers to doing so. For example, parents who deemed themselves low-risk were not willing to wear a face covering at home around their child/ren, even if their child/ren had tested positive for COVID-19. For several parents, engagement in protective behaviours was not compatible with caring for their child on a practical level. It should be noted, however, that the parents within the current sample were predominantly from non-vulnerable households and had had experience of living in the pandemic for 12 months or more by the time of their interview, so perhaps may not have shared the same attitudes towards protective behaviours as those in more vulnerable families, or even those they themselves may have held at an earlier stage of the pandemic. This finding regarding how risk perceptions relate to protective behaviours is very much in line with previous research that suggests risk perception, as well as perceived response efficacy of infection-control behaviours, is significantly related to the adoption of preventive health behaviours [36,37,38,39]. These findings also provide support for theoretical models such as the Health Belief Model [40] and Protection Motivation Theory [41]. These theories propose that an individual’s motivation to undertake a particular behaviour is in part influenced by their perceptions of risk or threat relating to that behaviour in the context of the current study, and their perceptions of their susceptibility to and severity of COVID-19.

Individuals’ perceptions of risk appeared to be influenced by several factors, such as whether they or someone in their household is at risk of serious illness from COVID-19, beliefs about within-home transmission, and the implementation of the vaccination programme. In addition, some school staff members were concerned that their students and/or their students’ families would not engage in infection-control behaviours, which they believed would increase their risk of contracting COVID-19 at school and reduce their control over their COVID-19 exposure. However, most school staff members reported feeling safer once they were vaccinated. These findings align with existing research that suggests that trust (i.e., confidence in those managing risk), visibility (i.e., how immediately evident a risk is), voluntariness (i.e., the extent to which a person has control over their exposure), and knowledge (i.e., the extent to which a person understands the risk) are key factors that generally alter risk perceptions [42]. More recent evidence suggests direct personal experience with COVID-19 and perceived harmfulness of the infection is related to higher risk perception [39,43]. As risk perceptions appeared so instrumental in determining the perceived acceptability and feasibility of recommended behaviours, it may be beneficial for future adaptations to Germ Defence to include features that support users to assess and understand their risk.

The intervention content was designed to promote a sense of shared responsibility, i.e., everyone collectively taking action to protect themselves and others more vulnerable against COVID-19. This feature was perceived by the majority of participants as a persuasive argument for adopting the suggested behaviours. Research suggests that the role of prosocial values is an important predictor of adopting protective health behaviours during the COVID-19 pandemic [38,39]. This finding highlights the benefit of encouraging students using Germ Defence to enact even small changes, collectively or individually, for reducing overall risk. In addition, our findings provide support for the sense of shared responsibility as a potentially important influence on behaviour. 

Our results also highlight several logistical barriers that sometimes make it difficult to implement the intervention’s recommended behaviours, in school and in the home. At school, these barriers were primarily related to the lack of physical space and appropriate resources. Previous research has also shown that having sufficient space is an important factor in the feasibility of socially distancing [26] and highlights an urgent need for alternative options for infection-control measures to be made available in school settings. Whilst a lack of physical space and appropriate resources was occasionally an issue within the home, the major practical barrier to infection-control behaviours was around the physical care needs of young children (e.g., feeding, dressing, personal hygiene, providing comfort). This finding mirrors what has been suggested previously regarding the importance of physical closeness between household members, even when a household is identified as high risk [36,44]. Although content was included in which these challenges were acknowledged, and users were encouraged to action what was possible, future research is needed to explore how such logistical barriers may be overcome to allow for optimal infection control in future pandemics.

Existing evidence has shown Germ Defence to be effective with various adult populations and acceptable in relation to COVID-19 [11,13,14,15,35]. The current study led to adaptations to the intervention for use in the school community and provides additional preliminary evidence that Germ Defence may be successfully adapted for use in different contexts and with different populations. Further research is now needed to explore the efficacy of the adapted content for increasing infection-control behaviours in schools.


**Strengths and limitations**


The breadth and depth of input from students, school staff, and parents allowed Germ Defence to be optimised for a variety of users. Co-designing interventions with young people is identified as a key strategy for enhancing engagement [45]. As such, it is hoped that our in-depth involvement of young people aged 11–18 at all stages of optimising Germ Defence will improve the likelihood of it being engaging for students. Additionally, the systematic and transparent process of adapting the intervention is a key strength of this study. The rigorous iterative process of modifying Germ Defence documented in the table of changes allowed the research team to efficiently make changes with a clear record of the decision-making process. These changes were closely informed by the target users’ experiences and knowledge at all stages. 

There are however a number of limitations. Recruitment challenges resulted in a small sample size consisting of a relatively homogenous group of school staff of largely female participants, who predominantly identified as English, Welsh, Scottish, Northern Irish, or British. Additionally, eight did not provide demographic information. Complete demographic data were also not collected from PPI members. As such, we are unable to fully describe the sample and so cannot be sure of the degree of diversity of the group to which our findings are relevant. An additional limitation relates to the (necessary) online nature of the study. Despite the think-aloud method being successfully used remotely in a recent study [15], it is plausible that we were not able to fully realise the benefits of this method due to the remote nature of the interviews. For instance, the interviewer was not always able to see the intervention content the interviewee was viewing, and consequently not able to ask follow-up questions to provoke further discussion. Nonetheless, this study does provide further evidence that the think-aloud method can be implemented remotely for the majority of participants. 

Due to the scope of the current study, the voices of primary-school-aged children along with those of staff, parents, and secondary school students from SEN schools are missing from this research. These groups are likely to have significantly different experiences of the pandemic compared to the sample reported here, and consequently may require different interventional support. Similarly, restrictions on time and funding meant that the school-adapted version of Germ Defence has not been translated into languages other than English, and therefore its reach is limited.


**Future directions**


The adapted Germ Defence for school communities aims to increase knowledge and implementation of infection-control behaviours, and as a result, to help reduce the transmission of COVID-19 among school students, staff, and parents. Future research should now be concerned with exploring how Germ Defence may be used to reduce transmission of infections outside pandemic settings. Indeed, since the underlying messages and principles of the original Germ Defence intervention were created to reduce transmission of all respiratory infections and were shown to be effective for this purpose and for reducing gastrointestinal infection [11], the school-adapted intervention may also support infection-control behaviours to reduce transmission of other infectious diseases, such as the common cold, flu, and stomach bugs within a community setting. Additionally, it may be advantageous to explore how Germ Defence can best be used within school communities (e.g., as a lesson/homework task, an assembly discussion or for parents to use within the home on a technological device) as well as to evaluate the real-life implementation of Germ Defence and its effects on behaviour and infections in school settings.

## 5. Conclusions

By adapting Germ Defence and optimising the intervention for school communities, we aimed to provide school students, staff members, and their families with an effective support resource to help mitigate against infection and transmission of COVID-19 within and from school settings. As well as providing a systematic overview of the process through which Germ Defence was adapted, our findings provide insight into which infection-control behaviours are broadly considered acceptable and feasible to implement amongst students, school staff, and parents, such as hand washing/sanitising and mask wearing outside the home. However, our findings also indicate that individuals’ perceptions of risk are important in deciding how feasible and acceptable behavioural recommendations are in this context, and that certain logistical barriers, including childcare needs and lack of physical space, make some behavioural recommendations unrealistic. Due to the rigorous methodological approach and breadth and depth of input from various stakeholders, Germ Defence for school communities offers an easily accessible and comprehensive source of support to combat against COVID-19 transmission that also has wider relevance for other infectious diseases. 

## Figures and Tables

**Figure 1 ijerph-19-06731-f001:**
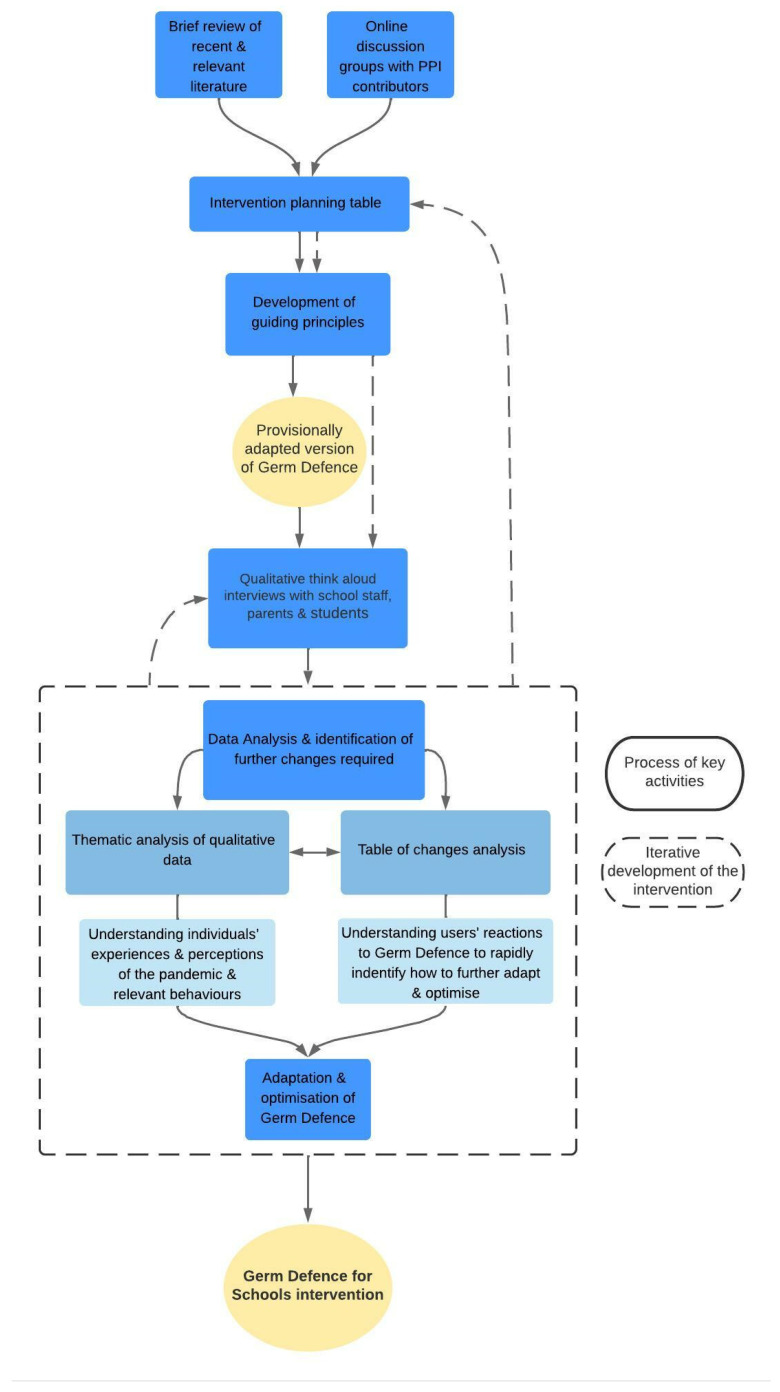
Key activities of the study design.

**Figure 2 ijerph-19-06731-f002:**
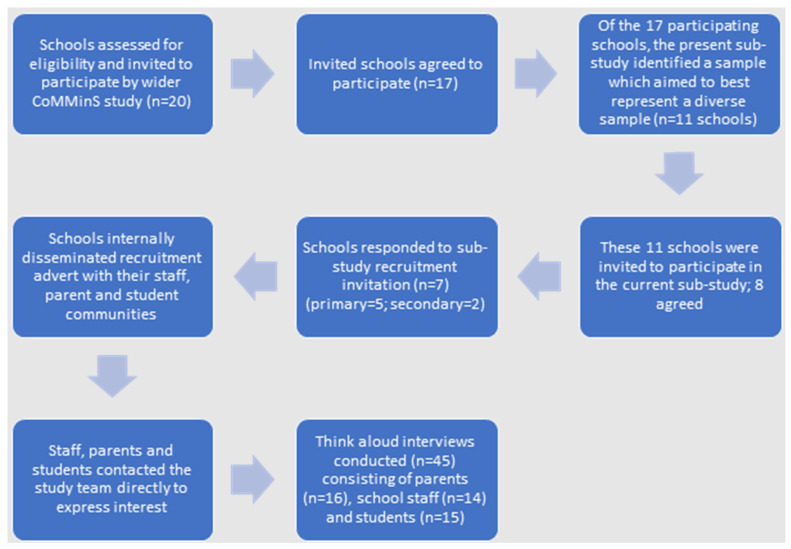
The study recruitment process.

**Table 1 ijerph-19-06731-t001:** Guiding principles for the optimised Germ Defence intervention.

User Context and Characteristics	Guiding Principle	
	Design Objective	Key Features
School staff already have lots of additional responsibility in the context of the pandemic and have very limited time/capacity to engage with online elements of intervention Students report limited attention span for such information provision; preference for ‘bitesize’ pieces of information presented in interactive ways where possible	Minimise work required to access and engage with Germ Defence—especially for school staff.	Simple, short pages with minimal text Optimise content and structure for mobile access Simple navigation and consistent page structure Separation of home and school sections to facilitate rapid identification of relevant information Align ‘home’ and ‘school’ sections to same modular design around each key behaviour Accompanying ‘How to Use’ guide for school staff
Many students and parents relatively unconcerned for their own risk from COVID but acknowledge others more at risk and need/desire to protect them Many students mention behaviour of peers when discussing behaviours they are/are not likely to adopt	Promote sense of collective responsibility for keeping ‘your community’ (i.e., home and family and/or school setting) safe.	Frame risk messages in terms of protecting vulnerable others and looking after whole community Use of social norms to encourage people (especially young people) to adhere to recommendations/not go along with peers who may not be doing so Highlight benefits of promoted actions for self AND others
Many parents and pupils express concern about others’ behaviours not adhering to recommendations and the sense that this negates their own efforts. Many school staff, parents and pupils recognise situations where certain measures are not possible to implement (e.g., not enough space in classrooms for everyone to stay 2 m apart)	Help users understand the objective of reducing rather than eliminating risk; reduce perceptions relating to measures not being worth doing if not being done perfectly.	Acknowledge difficulties in implementing all measures all the time and the need for adaptation to user context. Staged/stepped behavioural suggestions—starting with optimal solution and then alternatives if this not possible Acknowledge users’ current behaviours and provide persuasive suggestions as to how these can change. Provide encouragement for positive changes to planned, future behaviours.
Some confusion, particularly amongst students, about why some behaviours are required—especially in some circumstances but not others. School staff recognise that students generally implement behaviours well when they understand why they are being asked to do them Regularly changing/updating guidance sometimes leads to doubt about need for/importance of certain recommendations	Facilitate understanding of why each behavioural recommendation is important and how to overcome recognised barriers.	Present strong, accessible rationale for all behavioural recommendations (including explaining why recommendations change and vary) Acknowledge that measures may sometimes seem confusing/contradictory but explain in terms of risk reduction. Suggest strategies for managing situations/contexts where desired behaviours difficult/ not possible Encourage habit-forming behaviours by asking users to make a plan of future recommended behaviour. Explain when it is most important to perform key behaviours (e.g., wash/sanitise hands when coming in from outside, before eating, when touching shared objects, after coughing or sneezing, and after visiting the toilet). Provide a printable summary ‘poster’ to be displayed within the classroom/home providing reminders about key behaviours.
Widespread beliefs that if one person in the household is infected with COVID-19, inevitable that others will also get it	Persuade users that within-home transmission not inevitable and increase understanding of how this can be avoided.	Recommendations and strategies for implementing measures used outside the home within the home. Advice about dealing with visitors to the home and how to manage social expectations. Recommendations presented in format/structure that recognises that more stringent measures may only be feasible/acceptable in the home under certain circumstances/ in certain contexts.

**Table 2 ijerph-19-06731-t002:** Think-aloud interview participants’ demographic data.

	A School/College Student	A Member of School Staff	A Parent/Guardian of a School-Aged Child	A Member of School Staff and a Parent/Guardian of a School-Aged Child	All Participants
*n*	10	12	11	4	37
Age					
Range	12–17	25–62	32–47	34–41	12–62
Mean (SD)	14.4 (1.7)	39 (9.4)	40.7 (4.6)	38.8 (3.2)	32.8 (12.8)
Gender					
Male	6.7 (67.7%)	2 (16.7%)	2 (18.2%)		6.2 (16.7%)
Female	2.2 (22.2%)	10 (83.3%)	9 (81.8%)	4 (100%)	29.8 (80.6%)
Other	1.1 (11.1%)				1 (2.8%)
Ethnicity					
English/Welsh/Scottish/Northern Irish/British	5 (50%)	11 (91.7%)	10 (90.9%)	3 (75%)	29 (78.4%)
White and Asian		1 (8.3%)			1 (2.7%)
White and Black Caribbean	1 (10%)				1 (2.7%)
Other White	1 (10%)		1 (9.1%)		2 (5.4%)
Indian	1 (10%)			1 (25%)	2 (5.4%)
Other Asian	2 (20%)				2 (5.4%)
Highest Educational Level					
School		1 (8.3%)	1 (9.1%)		2.1 (5.6%)
College			2 (18.2%)	1 (25%)	3.1 (8.3%)
Undergraduate		5 (41.7%)	3 (27.3%)		8.2 (22.2%)
Postgraduate		6 (50%)	5 (45.5%)	3 (75%)	14.4 (38.9%)
Still in full-time education	10 (100%)				9.2 (25%)

**Table 3 ijerph-19-06731-t003:** Summary of main changes made.

Evidence	Examples	Subsequent Intervention Change
Many participants who deemed themselves/their household as ‘low risk’ reported some of the more stringent behavioural recommendations as unrealistic and reported themselves unlikely to adopt these.	*When I read the first paragraph I thought ‘Oh you’re joking!’ I don’t want to wear face coverings at home… I think it’s because it feels like a safe place because they’re horrible to wear and because I would think that because we don’t have people in the at-risk category in our household.* (P10) *It’s not always that practical… in our living arrangements. So if we, for example, have my elderly parents stay with us, space wise, I don’t think we could do that two metres apart indoors because we live in a small flat.* (P7) *There are some examples that you just need to quickly put one [face covering] on and it’s not really feasible.* (C15)	Addition of a high/low risk screening question/algorithm (to Home section) was added to display more ‘stringent’ behavioural planning questions only to users who deemed themselves at high risk. Reinforced the ‘try to do as much as possible’ message and added in wording such as ‘when and where possible’.
Common barriers to certain behavioural recommendations were identified: limited physical space (both in school and at home); needing to care for young children; lack of opportunity to always wash hands with soap and water at school; opening school windows leading to being uncomfortable inside.	*The school’s not exactly built for this…because when everyone leaves the lessons at the same time, all 215 people are in one corridor so they can’t really try and keep us two metres apart otherwise everyone’s going to be really late for their lessons.* (C6) *Within school you can’t actually physically do a two-metre kind of social distancing within the actual classroom ‘cause there’s not enough space with 30 children and often you have like three or four members of staff in the room as well.* (T4) *It would be very difficult for me to do it because my five-year old’s a very cuddly little boy and to tell him that he couldn’t see his mummy for seven days while she’s locked in a bedroom would be quite hard. If you’ve got young children and they’re used to having you there all the time, it would be very difficult. It kind of resonates that it’s the sensible thing to do, but whether it’s feasible for all families to do, it does give people the idea of ‘if you can do it, then do it’.* (P1) *I almost never wash my hands before and after putting it [face covering] on, because there just wasn’t the time…in the corridor going into the classroom, not all the classrooms have sinks anyway.* (C15) *We don’t give the students the opportunity to wash their hands regularly within school—they come into the classroom, they sanitise their hands and then, when they leave the classroom, they sanitise their hands, but they don’t wash them as such.* (T3) *So, the fresh air thing we are doing that but obviously in the winter that was a bit of a tricky situation because we were told we obviously had to shut the windows because it was absolutely freezing. So, that was a bit tricky.* (T8)	Reinforced the ‘do as much as possible’ message to acknowledge that there will be circumstances in which these are not possible. Suggestion to parents of wearing a face covering when with their child inside their home was still better than not doing so in a situation where a family member may be vulnerable and/or be infected. Included ‘hand sanitiser’ as an alternative within advice re handwashing. Added suggestions to overcome certain barriers experienced by students and school staff e.g., to bring in extra layers of clothing and to also open internal doors for ventilation.
Mixed feedback regarding the presentation of information about vaccinations; many participants deemed it important for students to be aware of, while others felt the information was not relevant to young people. N.B. During the development of this content it was unclear as to whether/ when young people would be offered vaccination.	*I think they need to know why it’s important for them to have a vaccination.* (T3) *It might just confuse them [children] thinking, ’Why am I reading a vaccination page?’ Or, ’Am I going to have a vaccination now?’ Or, ’Why am I reading it because I don’t think I get one.’ Sort of thing, so it’s a waste of time.* (P1) *I mean children maybe who were younger than me might not really find it that helpful, but people my age and definitely older people will find it really helpful.* (C2)	Added a message to the vaccinations page to clarify that, whilst the information may not currently be personally applicable to children, it was designed to help them understand how the vaccines may affect those around them, such as their family and teachers.
Several participants expressed confusion towards the questions regarding their current and planned behaviours. Many reported these pages looked too similar, and participants were unclear as to what they were being asked to do.	*On the ‘what would you do in the future’ [page] I think maybe you could simplify the way it’s explaining what you have to do for that bit? Because I was confused in what I actually have to do.* (C1) *I thought, why are they asking me it again? And then the second one was, now you’ve done this, what will you do better?* (T10)	Instructions on these pages were made clearer via rewording and displayed in a text box. Each of the pages were changed to have different-coloured backgrounds and key instructional text was highlighted.
Almost all participants described the intervention’s content as ‘too text-heavy’ stating that it also needed more colour and icons throughout to be engaging for all users, particularly children. Several participants stated that rewording jargon (e.g., ‘not applicable’ and ‘vulnerable’) was essential to improving accessibility.	*A few more pictures, maybe like a bit more colour, but it’s relatively informative and yeah, just to be made more to a child’s level of how they would speak rather than adults* (P6) *Sometimes when things are too long, people will just not read it and just skim past it, but if it’s like a short sentence in a bright green box, you’re more likely to just quickly read it* (C12) *I didn’t know what that [not applicable] means so maybe some other kids my age might not know what it means, or even older* (C7)	The amount of text was reduced and reworded to be more accessible with less jargon. Pages were better presented through the use of bullet points, text boxes and icons. Additional information was displayed via ‘dropdowns’.

## Data Availability

The data presented in this study are available on request from the University of Bristol Data Repository.

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
