# Peer review of "The Rapid Adaptation and Optimisation of a Digital Behaviour-Change Intervention to Reduce the Spread of COVID-19 in Schools"

_ijerph, 2022, doi:10.3390/ijerph19116731_

Round 1

Reviewer 1 Report

Dear authors,

this article is very interesting, and it can be published after the minor correction mentioned below:

1) In the introduction, please explain better the Germe Defence Approach: the guideline principles and messages.

2) Explain the abbreviations: PPI and RE&SD

3) Figure 1. Why the arrows have a different formatting (dashed and not dashed)? If they have a different meaning it must be explained below, otherwise they should have the same formatting

4) Add the small sample size as limitation

Author Response

Thank you for your feedback. 

In the introduction, we have added some text to better explain the approach used within Germ Defence. Please see this text via tracked changes.

At the beginning of the paper on line 11, we have defined 'RE' and 'SD'. On line 118 and 119, we have defined what 'PPI' is (patient and public involvement). Please see these edits via tracked changes.  

Figure 1 is a flowchart showing the process of the intervention development (the key activities). This explanation of the non-dashed lines/arrows has been added to the figure’s key. We hope this helps.

We have added the small sample size as a limitation of the study within the discussion. Please see this text via tracked changes.

Reviewer 2 Report

The paper is focused on rapid adaptation and optimisation of a digital behaviour  change intervention to reduce the spread of COVID-19 in schools an interesting integrated approach.

The approach is well explained and the example allows readers to understand the whole problem, and also to replicate it.

This paper is a well-argued and valuable contribution by analysing experienced topic.

The paper is clearly structured, and contains all the necessary elements.  

References to modify the requirements of the magazine.

I recommend concluding. 

Author Response

Thank you for your feedback. We are happy to hear such positive comments.

Unfortunately, we do not understand the following comments:

  1. References to modify the requirements of the magazine. 
  2. I recommend concluding.

We feel that we have included all of the necessary references throughout the paper. We have also provided a conclusion, starting on line 837, at the end of the paper. Please let us know if you would like us to make further changes. 

Reviewer 3 Report

This paper presents a comprehensive analysis of the intervention 'Germ Defence'. I enjoyed reading it. The tables and figures are very helpful. I checked out the 'Germ Defence' website, and it is very interesting. It will be helpful to include a section to briefly talk about how the website was developed and the impact of this research on the website design. 

Author Response

Thank you for your feedback. 

We have added a small paragraph within the methods section to explain how the website was developed and the impact of our research on the design. Please see these edits via tracked changes. 

Round 2

Reviewer 2 Report

Dear autors,
comments were accepted.